# The Landscape of Noncoding RNA in Pulmonary Hypertension

**DOI:** 10.3390/biom12060796

**Published:** 2022-06-07

**Authors:** Lin Deng, Xiaofeng Han, Ziping Wang, Xiaowei Nie, Jinsong Bian

**Affiliations:** 1Department of Pharmacology, School of Medicine, Southern University of Science and Technology, Shenzhen 518055, China; dengl6@sustech.edu.cn (L.D.); 12133154@mail.sustech.edu.cn (Z.W.); 2Department of Diagnostic and Interventional Radiology, Beijing Anzhen Hospital, Capital Medical University, Beijing 100029, China; hanxiaofengaz@126.com; 3Shenzhen Key Laboratory of Respiratory Diseases, Shenzhen People’s Hospital (The First Affiliated Hospital, Southern University of Science and Technology), Shenzhen 518055, China

**Keywords:** pulmonary hypertension, noncoding RNA, long noncoding RNA, circular RNA, therapeutics

## Abstract

The transcriptome of pulmonary hypertension (PH) is complex and highly genetically heterogeneous, with noncoding RNA transcripts playing crucial roles. The majority of RNAs in the noncoding transcriptome are long noncoding RNAs (lncRNAs) with less circular RNAs (circRNAs), which are two characteristics gaining increasing attention in the forefront of RNA research field. These noncoding transcripts (especially lncRNAs and circRNAs) exert important regulatory functions in PH and emerge as potential disease biomarkers and therapeutic targets. Recent technological advancements have established great momentum for discovery and functional characterization of ncRNAs, which include broad transcriptome sequencing such as bulk RNA-sequence, single-cell and spatial transcriptomics, and RNA-protein/RNA interactions. In this review, we summarize the current research on the classification, biogenesis, and the biological functions and molecular mechanisms of these noncoding RNAs (ncRNAs) involved in the pulmonary vascular remodeling in PH. Furthermore, we highlight the utility and challenges of using these ncRNAs as biomarkers and therapeutics in PH.

## 1. The Noncoding RNA Transcriptome in Pulmonary Hypertension

The majority of the human genome has been transcribed but only ~2% of the genome contains protein-coding genes. The transcribed RNAs with low/no protein coding potential are collectively referred as non-coding RNAs (ncRNAs), including small and long noncoding RNAs (sncRNAs and lncRNAs, respectively) and are historically classified by mature transcript length based on a 200-nucletotide (nt) cutoff. The sncRNAs include microRNAs (miRNAs), small interfering RNAs (siRNAs) and Piwi-interacting RNAs (piRNAs) [1]. MiRNAs are ~22 nt sncRNAs in length and are the most widely studied sncRNAs in human diseases, including pulmonary hypertension [2]. PiRNAs are newly discovered sncRNAs approximately 24–32 nt with an uracil bias at the 5′ end and 2′-O-methylation at 3′ end, which accomplish their regulatory functions through specifically binding to the PIWI subfamily of Argonaute proteins [3]. There are about 1.3 million A*lu* elements in human genome, only a handful of which can amplify through RNA Pol III [4]. RNA Pol III synthesized A*lu* RNAs are generally expressed at very low levels in cell, and there expression levels can increase under pathological conditions [5]. A*lu* RNAs can be transcribed in a variety of contexts including introns, mature mRNAs and non-coding RNA transcripts. Studies have found that A*lu* RNAs are involved in translational and transcriptional mechanisms [6]. lncRNAs are heterogeneous noncoding RNA transcripts longer than 200 nt in size. Based on genomic localization, lncRNAs can be classified into subgroups including Sense, Intronic, Antisense, Bidirectional, and Intergenic lncRNAs [7]. In addition, circular RNAs (circRNAs), a class of highly abundant and evolutionary conserved ncRNAs, are covalently closed RNA molecules that lacks free 3′ and 5′ ends and have recently been identified as functional RNA molecules [8,9]. In addition to their regulatory roles as noncoding transcripts, evidence has recognized that a small portion of lncRNAs and circRNAs contain short open reading frames (sORFs) that can be translated into functional small peptides [10].

Pulmonary hypertension (PH) is a progressive cardiopulmonary disease characterized by small pulmonary vascular remodeling/plexiform lesions. Elevated mean pulmonary arterial pressure results in right ventricular hypertrophy and failure, which is the most common cause of death in PH patients [11]. There are many ncRNAs aberrantly expressed in PH patients and animal PH models. Numerous studies have shown that the regulatory roles of ncRNAs are involved in the pathogenesis of PH, including pulmonary vascular remodeling and right ventricular remodeling (Figure 1). In particular, miRNAs have been extensively investigated in PH (Reviewed by [2,12,13]). Although there are no miRNA-based therapeutics under clinical trials in PH, preclinical models have shown beneficial effects of miRNA-based therapy. For example, miRNAs such as miR-29, miR-124, miR-140 and miR-204, are considered good candidates for therapeutic targets, as they have shown consistent pattern of expression profiles in PAH patients and different in vivo and ex vivo experimental PH models [14,15]. In addition, studies have looked at the correlation between the expression profiles of dysregulated miRNAs and the disease severity or survival rate of pulmonary arterial hypertension (PAH) patients. Multiple promising miRNAs have been considered as biomarker of PAH disease that are being actively evaluated [16,17]. Aside from miRNAs, lncRNAs and circRNAs are other two major groups of ncRNAs relatively less investigated and understood in the PH field. However, they are attracting growing attention due to their abundance, regulatory functional roles, cell/tissue expression specificity in human diseases, and potential therapeutic targets and disease biomarkers [18,19,20,21]. This review focuses on the recent developments of lncRNA and circRNA related to PH.

## 2. Long Noncoding RNAs (LncRNAs) in Pulmonary Hypertension

Before the development of high-throughput sequence technologies, the non-coding genome was considered as transcriptional noise or junk DNA. Over the last decades, a large number of transcriptomic studies with unprecedented resolution and scale in the noncoding genome have revealed that lncRNAs are one of the most pervasive noncoding RNAs in the human transcriptome, subsequently attracting increasing attention and driving functional investigation in the pathophysiology of human diseases. The recent updated human GENCODE version GENCODE 35 (August 2020) identified 17,957 lncRNA genes and 48,684 transcripts [22]. With the development of new technologies in sequencing and bioinformatics analysis pipelines, as well as transcriptome investigations in additional cells/tissues/organs, more cell/tissue-expressed LncRNAs and transiently expressed lncRNAs will be identified. However, the functional and regulatory mechanisms of the majority of lncRNAs in human diseases are still largely unknown. These large number of lncRNAs in the human transcriptome have resulted in increasing scientific interest and leading-edge research in human diseases. Functional and mechanistic investigation of lncRNAs will provide better understanding of the pathogenesis of diseases (including pulmonary hypertension), which will be eventually contribute to the development of lncRNA-based novel therapeutics and biomarkers (diagnostic, predictive, prognostic, and therapeutic).

### 2.1. LncRNA Biogenesis and Function

The biogenesis of lncRNAs is similar to that of protein-coding mRNAs, which are transcribed by RNA Pol II from independent promoters and can undergo post-transcriptional splicing and modification in the nucleus. Most lncRNAs contain 5′-end m^7^ G caps and 3′-end ploy (A) tails similar to mRNAs. However, there are distinct roles of lncRNA regarding to transcription, processing, export and turnover, which are closely associated with their cellular functions and fates [20]. lncRNAs contain fewer and longer exons and the splicing efficiency is lower than in mRNAs. The expression levels of lncRNAs are relatively low and less evolutionarily conserved compared with mRNAs and miRNAs [23,24,25]. LncRNAs can be divided into different groups based on the transcribed regions in the genome: lincRNA (long intergenic ncRNA), intronic lncRNA, eRNAs (enhancer RNAs), sense or antisense (sense or antisense to another gene), and bidirectional [26]. Unlike mRNAs, lncRNAs exhibit diverse subcellular distribution patterns, ranging from the majority of lncRNAs with a predominantly nuclear localization, and fewer lncRNAs exclusively with cytoplasmic localizations, as well as those with both cytoplasm and nucleus localization. The subcellular localization of lncRNAs is critical for their biological functions and cellular behaviour [27]. Generally, nuclear lncRNAs function as regulators to modulate gene expression in *cis* or *trans* at the epigenetic and transcriptional level via signals, decoys, guides, scaffolds and enhancers [28]. Cytoplasmic lncRNAs can modulate mRNA stability, translational or posttranscriptional control of gene expression and signal transduction pathways. One of the key mechanisms is that lncRNAs can act as competitive endogenous RNAs (ceRNAs) to achieve regulation of mRNA [29], which can block miRNA activity through sequestration, thereby derepressing the miRNA targets [30] and participating in the regulation of cellular biological processes [31].

Many studies have demonstrated that lncRNAs are involved in numerous important biological processes. The gain-of-function and loss-of-function of lncRNAs have been implicated in human diseases such as cardiovascular diseases [32,33]. LncRNAs exert their function as guide, decoy, scaffold, signal, sponge or ceRNA through interacting with DNA, RNA and proteins [31]. In addition, lncRNA can work in *cis* (local) on regulating neighboring genes or in *trans* (distal), regulating genes distantly, or as molecular targets at the transcriptional and post-transcriptional levels in the nucleus or cytoplasm [34,35]. Given the functional and mechanistic diversity of lncRNAs, it is not surprising that many of the lncRNAs have been found to be important contributors in PH disease progression and represent promising therapeutic targets.

### 2.2. LncRNAs in Pulmonary Artery Endothelial Cells in PH

Vascular endothelial cells (ECs) are located at the innermost layer of blood vessels and are directly exposed to blood flow and various stimuli [36]. In PH, endothelial dysfunction is the critical initiation event that triggers the development of disease [37]. The apoptosis of EC induced by a pathogenic stimulus result in the degeneration of the fragile distal pulmonary arterioles and leads to further microvascular loss. First, the apoptosis of EC releases mediators that promote smooth muscle cell proliferation. Second, the persistent EC apoptosis may lead to antiapoptotic signaling activation and selection of apoptosis-resistant and pro-proliferation ECs [38,39]. Thus, in PH, the initial apoptosis of distal pulmonary artery endothelial cells (PAECs) leads to the selection of hyperproliferative and pathogenic PAECs that stimulate cell proliferation, migration and resistance to apoptosis in the intima, media and adventitia layers of pulmonary arteries resulting in the later stage of pulmonary vascular remodeling and plexiform lesion formation.

Little is known about the molecular mechanisms and biological functions of lncRNA in the endothelial cell in the pathogenesis of PH. Microarray analysis performed on endothelial tissues from the pulmonary arteries of chronic theromboembolic pulmonary hypertension (CTEPH) patients and healthy controls identified hundreds of dysregulated lncRNAs associated with disease progression. Gene ontology and pathway analysis showed that the differentially expressed lncRNAs were involved in the regulation processes of the inflammatory response, response to endogenous stimuli and antigen processing and presentation [40]. It is well documented that chronic hypoxia drives pulmonary vascular remodeling and right ventricular hypertrophy [41]. Bulk RNA-Seq performed in human pulmonary microvascular endothelial cells (PMECs), pulmonary artery smooth muscle cells (PASMCs) and pericytes exposure to a hypoxic condition identified numerous novel lncRNAs, revealing 251 (162 upregulated), 275 (140 upregulated), and 290 (176 upregulated) of dysregulated lncRNAs, respectively. LncRNA LNCOG and LncRNA TUG1 are only two lncRNA differentially expressed among three cell types, but with inconsistent patterns, which is consistent with cell-specific regulation of lncRNAs. Further gene ontology (GO) and KEGG pathway analysis showed that PMECs have unique regulation patterns, including the intrinsic apoptotic signaling pathway response to oxidative stress, the mTOR signaling pathway and cell senescence [42] This study comprehensively analyzed the lncRNA transcriptomes of pulmonary vascular cells and pericytes exposure to hypoxia and revealed distinctive regulation patterns of individual lncRNA. The tissue-specific and cell-specific expression of lncRNA in PH disease suggest that dysregulated individual lncRNAs are promising disease biomarkers and therapeutic targets. LncRNA MANTIS is highly expressed in endothelial cells compared with other cell types. MANTIS expression has seen to be decreased in the lung tissues from IPAH patients and Rat MCT-PH model. However, MANTIS expression was upregulated in the carotid arteries of *Macaca fascicularis* subjected to an atherosclerosis regression diet, and in endothelial cells isolated from human glioblastoma patients. These inconsistent expression patterns suggest disease-specific expression of lncRNA MANTIS. Silence of MANTIS inhibited PAEC angiogenesis by a tube formation assay, which may explain the disruption of angiogenesis in the distal pulmonary arterials of PAH patients due to downregulation of MANTIS [43].

It is well known that a heterogeneous population of endothelial cells exists in various vascular beds. Endothelial cell heterogeneity plays a vital role in the pathogenesis of PH [44]. During the development of PH, certain EC populations undergo a process called endothelial-to-mesenchymal transition (EndMT), which was observed in the remodeled pulmonary arteries and right ventricular vessels of PAH patients and rodent PH models [45,46,47,48]. EndMT is a biological process in which endothelial cells lose their specific phenotype and gain a mesenchymal cell phenotype (e.g., smooth muscle, fibroblast, and myofibroblast-like cells) [49]. LncRNA MALAT1 was the first lncRNA identified as being upregulated in TGF-β1-induced EndMT in endothelial progenitor cells (EPCs). Manipulation of MALAT1 modulates the EndMT process by regulating TGFBR2 and SMAD3 by directly binding with miR-145 [50]. Further studies have strengthened the important role of MALAT1 in regulating the EndMT process, in which inhibition of MALAT1 prevents the high glucose-induced EndMT by regulating the miR-205-5p/VEGF-A axis [51]. In addition, upregulation of MALAT1 facilitates the oxidized low-density lipoprotein (ox-LDL) induced EndMT through Wnt/β-catenin signaling pathway [52]. In the same EndMT in vitro model induced by Ox-LDL [53], LINC00657 promotes EndMT by acting as a ceRNA targeting miR-30c-5p and by activation of the Wnt7/β-catenin signaling pathway [54]. LncRNA ZFAS1 triggers EndMT by targeting the miR-150-5p/Notch3 axis in an atherosclerosis model both in vivo and in vitro [55]. The upregulation of ox-LDL was found in PAH patients [56]. However, whether ox-LDL can induce EndMT in PH still unknown.

One of the key mechanisms of lncRNA regulation of gene expression is that lncRNA act as “ceRNAs” sponging miRNA and interfering with miRNA function, thereby derepressing the mRNA targeted by the miRNA [57]. For instance, lncRNA TUG1 promotes the LPS-mediated EndMT process in liver sinusoidal endothelial cells (LSECs), inducing ATG5 expression by sponging miR-142-3p [58]. LncRNA UCA1 promoted TGF-β1-induced EndMT in human umbilical vein endothelial cells (HUVECs), and silencing of UCA1 prevented the EndMT process through the UCA1-miR-455-ZEB1 pathway [59]. Mesenchymal stem cells (MSCs)-derived exosomal lncRNA SNHG7 prevented high glucose-induced EndMT by sponging miR-34a-5p [60]. Many studies have shown that lncRNAs function as ceRNA in the pathogenesis of PH, but their functional role in the regulation of EndMT in PH is still unknown.

Additional lncRNAs have been implicated in the regulation of EndMT. H19 knockdown significantly attenuated kidney fibrosis by inhibition of EndMT [61]. H19 knockout inhibited the EndMT process of pulmonary microvascular ECs in a streptozotocin (STZ)-induced hyperglycemia mouse model by H19/TE1 signaling [62]. PM2.5 is one of the major pollutants during our daily life. PM2.5 exposure can aggravate and induce PH disease [63]. Mouse chronic exposure to PM2.5 induced the EndMT process in the lung tissues. A set of lncRNAs associated with EndMT were identified by lncRNA microarray on the lung tissues from control and PM2.5-exposed mice. The downregulated LncRNA Gm16410 was selected for further functional and mechanistic study, whose overexpression prevented the EndMT process through the TGF-β/Smad3/p-Smad3 pathway [64]. LncRNA MEG3 overexpression suppressed EndMT in both in vitro and in vivo (diabetic retinopathy rat model) by inhibition of the PI3K/Akt/mTOR signaling pathway [65]. Antisense lncRNA GATA6-AS was found to facilitate the EndMT process, and silencing GATA6-AS prevented TGF-β2-induced EndMT by binding with LOXL2 (Lysyl oxidase homolog 2) to regulate endothelial gene expression through chromatin remodeling [66].

In pulmonary arterial hypertension (PAH), bulk RNA-Seq and single RNA-Seq were performed on the PAECs and HUVECs EndMT models induced by co-stimulation with TGF-β2 and IL-1β. The transcriptional architecture of lncRNAs relevant to EndMT was first identified in the PAH field. The downregulation of selected lncRNA MIR503HG was further validated in BOECs and remodeled pulmonary arteries from PAH patients as well as in mouse PH lung tissues, which suggests the pathological relevance of MIR503HG regulation of EndMT in PAH. Silencing of MIR-503HG induced a spontaneous EndMT phenotype, which suggests that the decreased expression of MIR503HG contributed to the EndMT process. Importantly, overexpression of MIR503HG repressed the EndMT process in both in vitro and in vivo mouse PH models, partially through interacting with PTBP1 (Polypyrimidine Tract Binding Protein 1) [67].

The above studies investigated the role of lncRNA regulating EndMT process in multiple diseases including PH (MIR503HG). However, most research has involved lack of functional and mechanistic studies, in vivo validation and clinical sample analysis. The reasons may be lncRNAs poorly conserved across species and their relatively low expression. To date, the role of EndMT related lncRNAs in human disease is still largely unknown. As the human genome contains large numbers of cell-specific and tissue-specific lncRNAs (e.g., endothelial cell specific lnRNA), we expect a fast-growing field of research in lncRNAs associated EndMT in the future.

### 2.3. LncRNA in Pulmonary Artery Smooth Muscle Cells in PH

Currently available PH therapies predominantly correct endothelial dysfunction by inhibiting the endothelin pathway and enhancing the prostacyclin (PGI_2_) and NO pathways to improve symptoms and survival. However, no available treatment can reverse the pulmonary vascular remodeling process during the development of PH, which is the pathological hallmark of PH [68]. Pulmonary endothelial cell dysfunction and damage play a vital role in the initiation of PH disease. EC apoptosis and damage can release several endothelial-derived factors and active cellular signaling pathways through autocrine, paracrine and endocrine functions, which lead to pulmonary arterial microenvironment dysregulation contributing to the pulmonary vascular remodeling [69]. Excessive PASMC (a major component of the pulmonary arterial wall) proliferation, apoptosis resistance and increased migration potential contribute to distal pulmonary artery remodeling and plexiform lesions formation. In addition, the remarkable ability of PASMCs dynamically modulate their contractile and synthetic functional phenotypes, indicating their central role in the physiopathological process of pulmonary vascular remodeling [70]. The functional and mechanistic role of PASMCs in the development of PH is still largely unknown. Only few lncRNAs have been investigated in endothelial cells of PH, while the majority of lncRNA studies have concentrated on their regulatory role of the PASMCs during the development of PH [33,71,72,73]. LncRNA functionality is dependent on subcellular localization, which is critical in understanding lncRNA interaction partners, mechanism and translational potential [27]. As not all the studies analyzed the subcellular localization of lncRNAs in PASMCs, here we only summarize lncRNAs with known cellular distributions in PASMCs.

### 2.4. LncRNA Distributed in Cytoplasm and Functional as CeRNA in PASMCs

An LncRNA can serve as an endogenous “sponge” to regulate mRNAs by targeting miRNA, which is based on the ceRNA theory. The establishment of intricate lncRNA-miRNA-mRNA coregulation networks have shown wide relevance in PH. LncRNAs have multiple target miRNAs, and miRNAs have several (even hundreds) of target mRNAs, that contribute to the complexity of networks. The further development of computational tools and wet lab experimental techniques that reveal the intricate connections existing in the lncRNA-miRNA-mRNA networks is becoming necessary and would add significant value to functional characterization and understanding molecular mechanisms.

LncRNA maternally expressed gene 3 (MEG3) is a well-known tumor suppressor lncRNA [74] that is primarily localized in the cytoplasm of PASMCs [75]. MEG3 expression was decreased in the total lung and pulmonary arteries of PAH patients. Silencing of MEG3 promoted PASMC proliferation by regulating cell cycles progression through PCNA, Cyclin A and Cyclin E protein expression. In addition, MGE3 could activate p53 signaling by affecting p53 nuclear translocation [76]. An independent study showed opposite results, in that MEG3 expression was upregulated in hypoxia-induced PASMCs and PASMCs-derived from a PAH patient. Silencing of MEG3 inhibited proliferation and cell-cycle progression though the miR-328-3p/insulin-like growth factor 1 (IGF1R) axis. Importantly, local delivery of siRNA MEG3 in vivo significantly prevented the development of hypoxia-induced PH in mice [75], suggesting the therapeutic potential of targeting MEG3. Due to the inconsistent findings of MEG3 in PH, further studies should address the following aspects: validate the expression profiles of MEG3 in independent PH patient cohorts and different animal PH models; generate genetic knockout or transgenic mice to access the PH phenotype, and investigate the effect of pharmacological inhibition of MEG3 by GapmeR and other inhibition strategies on mice or rat PH models.

Most lncRNAs associated with PASMC function in PH were initially screened by expression validation in rodent models or hypoxia-induced PASMCs. Growth arrest-specific 5 (GAS5) expression was decreased in a hypoxia-induced rat PH model, and hypoxia induced PASMCs. Silencing of GAS5 promoted the proliferation and migration of PASMCs through the miR-23b-3p/KCNK3 axis [77]. Consistent with this finding, downregulation of GAS5 was found in PASMCs treated with PDGF-BB. Manipulation of GAS5 affected the viability and migration of PASMCs in vitro. Overexpression or knockdown of GAS5 by rats subcutaneously inoculated with PASMCs transfected with GAS5 overexpression vector or shGAS5 vector in vivo, modulated pulmonary artery wall thickening, angiogenesis and autophagy in a chronic thromboembolic pulmonary hypertension (CTEPH) rat model by targeting miR-382-3p [78]. However, the translational potential of GAS5 in PH is still a matter for extensive, future research.

LncRNA MALAT1 expression was increased in PASMCs with hypoxic exposure based on lncRNA microarray data, which may be regulated by the hypoxia-inducible factor 1α. Knockdown of MALAT1 significantly decreased PASMC proliferation and migration in vitro and alleviated the right ventricular hypertrophy in a mouse PH model [79]. Pulmonary arteries, PASMCs and plasma derived from PAH patients further validated the upregulation of MALAT1 in PAH disease. Manipulation of MALAT1 affected PASMCs proliferation, migration, apoptosis, and the cell cycle through the miR-124-3p.1/KLF5 or miR-503/TLR4 axis [80,81].

LncRNA CASC2 was downregulated in pulmonary arteries in a hypoxia-induced rat PH model and hypoxic PASMCs. Overexpression of CASC2 inhibited cell proliferation, migration, apoptosis resistance and pulmonary vascular remodeling in a hypoxia-induced rat PH model [82]. An independent study confirmed the decrease of CASC2 in hypoxia-induced PASMCs and plasma samples from PAH patients and healthy controls. Mechanically, CASC2 regulates cellular function through the miR-222/ING5 axis, which was dysregulated in PAH disease with increase expression of miR-222 and decreased expression of ING5 in patient plasma samples [83].

Similarly, lncRNA TUG1 expression was increased in hypoxic PASMCs and total lung tissues from PAH patients in a hypoxia-induced PH model [42,84,85]. The cellular distribution of TUG1 is localized in both the nucleus and cytoplasm. Silencing of TUG1 attenuated cell proliferation, migration and apoptosis resistance, as well as cell-cycle progression in PASMCs [84,85]. In addition, inhibition of TUG1 in vivo significantly reduced right ventricular pressure, right ventricular hypertrophy and pulmonary vascular remodeling [84,85]. The mechanisms involved in the regulation of TUG1 in PH disease include interacting with miR-328 [85], sponging the miR-374c/Foxc1 axis [84] and regulating the miR-145-5p/SOX/BMP axis, miR-129-5p/CYP1B1/VCP axis and miR-138-5p/KCNK3/RHOC axis [42]. lncRNAs including NEAT1 [86], AC068039.4 s [87], LincRNA-COX2 s [88], Lnc-Ang362 [89] and SMILR [90], have increased expression in hypoxic PASMCs and clinical samples from PAH patients. Knockdown of these lncRNAs inhibited proliferation and migration and induced apoptosis via different mechanisms including sponging the miR-34a-5p/KLF axis [86], miR-26-5p/TRPC6 axis [87], miR-let-7a/STAT3 axis [88], miR-221/222 [89], and the miR-141/RhoA/ROCK signaling pathway, [90] separately. Furthermore, inhibition of SMILR prevented the development of PH by inhibiting PASMC proliferation and pulmonary vascular remodeling [90]. In addition, H19 expression was increased in the serum and lung tissues of an MCT PH model, and PDGF-BB stimulated PASMCs. Overexpression of H19 induced cell proliferation through the let-7b/AT1R axis, and knockout of H19 protected the development of PH [91]. LncRNA PAHRF expression was decreased in the pulmonary arteries of PAH patients and hypoxic PASMCs. Overexpression of PAHRF inhibited PASMC proliferation through the miR-23a-3p/MST1 axis [92]. Taken together, as described above, cytoplasmic lncRNAs can target miRNAs to reduce their regulatory effect on mRNAs participating in the pathogenesis of PH. Theoretically, one lncRNA can have multiple binding sites for a single miRNA, and various binding sites for numerous miRNAs. One single miRNA can target large amounts of mRNA. Collectively, the lnRNA-miRNA-mRNA axis seems to be quite complex, and the underlying molecular mechanisms are still largely unknown.

### 2.5. LncRNA Distributed in Both Nucleus and Cytoplasm in PASMCs

LncRNA Rps4l expression was decreased in lung tissues in a hypoxia-induced mouse PH model from bulk RNA-Seq, and its downregulation was further validated in hypoxic PASMCs. Rps4l can regulate cell functions including proliferation, migration and cell cycle progression through the regulation of interleukin enhancer-binding factor 3 (ILF3) and hypoxia-inducible factor 1 (HIF-1a) in PASMCs in *trans*. Transgenic mouse overexpression of Rps4l prevented the development of PH by reducing pulmonary vascular remodeling and right ventricular hypertrophy [93]. Further study found that RPS4XL, a novel peptide encoded by Rps4l, was downregulated in hypoxia-induced PH and hypoxic PASMCs. RPS4XL prevented hypoxia-induced PASMC proliferation by inhibition of the phosphorylation of its binding protein, RPS6. Manipulation of Rps4l by AAV9 overexpression showed a protective role in a mouse PH model, which was consistent with the protective effect of Rps4l in transgenic mice. Interestingly, mutation of the ORF of Rps4l caused a loss of the therapeutic effect, which suggests that the mini peptide RPS4XL is the main functional part of lncRNA Rps4l. Mechanistically, the protective effect of RPS4XL in PH is mediated through inhibition of RPS6, silencing of which prevented the development of PH [94]. Due to the poor conservation of lncRNA across species, whether lncRNA Rps4l has human homologs is still not yet clear. Although the mouse PH model showed the promising therapeutic effect of Rps4l, the clinical translation of Rps4l is still unsure. Future studies need to confirm the conservation pattern of Rps4l, and the expression profile in the pulmonary vasculature and right heart tissues. If there is no human homologue of Rps4l, investigation of the interaction targets of Rps4l in a rodent PH model, human pulmonary vascular cells and clinical samples of PAH patients would be optimal strategies.

Conserved lncRNAs have great potential for clinical translation. LncRNA Hoxa-AS3 expression was increased in pulmonary arteries from hypoxia-induced and MCT-induced mice in a PH mouse model, and upregulation was further confirmed in the PASMCs derived from PAH patients and hypoxia-induced PASMCs. In situ hybridization showed that Hoxa-AS3 was highly expressed in the smooth muscle layer. The dysregulated expression of Hoxa-AS3 in PASMCs under hypoxia exposure may be associated with H3K9 acetylation and HIF-1a. Gain-of-function (GoF) and loss-of-function (LoF) of Hoxa-AS3 affected cell proliferation and the cell cycle program through *cis*-regulation of nearby gene Hoxa3 [95]. Hoxa-AS3 exhibits a dual nuclear and cytoplasmic distribution in PASMCs. Apart from its regulating gene in *cis* as a nuclear lncRNA, Hoxa-AS3 can function as a ceRNA to sponge miR-675-3p by derepressing PDE5A expression in PASMCs [96].

A screen of novel lncRNAs from PAH clinical samples may discover strong PAH disease-related lncRNAs. Transcriptional profiling of lncRNA in PAH was performed by RNA-Seq of small remodeled pulmonary arteries from a large patient cohort including 18 IPAH patients and 17 healthy controls. PAXIP1-AS1 was selected from a set of differentially regulated lncRNAs due to its strongly dysregulation in PASMCs and fibroblasts. Silencing of PAXIP1-AS1 by siRNA or GapmeR inhibited the proliferation, migration and apoptosis resistance of PASMCs by targeting paxillin and FAK expression levels [97]. Independent research showed that PAXIP1-AS1 expression was significantly increased in a hypoxia-induced PH rat model and in hypoxic-PASMCs. Knockdown of PAXIP1-AS1 suppressed the hypoxia-induced cell viability and migration of PASMCs with a different mechanism, in which PAXIP1-AS1 act as scaffolds by interacting with WIPF1/RhoA complex and recruiting transcriptional factor ETS1 to form a regulatory complex. In addition, shRNA-mediated knockdown of PAXIP1-AS1 in vivo prevented the development of disease in a rat PH model [98]. RNA-Seq of human PASMCs and lung pericytes exposed to hypoxia and derived from IPAH patients revealed the transcriptome profiles of lncRNA associated with hyperproliferative and apoptosis-resistant phenotypes in PAH disease. Bioinformatic analysis identified that lncRNA TYKRIL (tyrosine kinase receptor-inducing long noncoding RNA) was the only commonly upregulated lncRNA in all conditions. The upregulation of TYKRIL was induced by pro-PH factors including PDGF, IL-18 and TGF-β, and was regulated by HIF-1α. Silencing of TYKRIL prevented cell proliferation and induced apoptosis in both hypoxia and PAH pathological conditions. RNA-Seq was performed on human pericytes with TYKRIL silencing and showed that the tyrosine kinase receptor PDGFRβ was significantly downregulated. GoF and LoF of TYKRIL demonstrated that TYKRIL can regulate PDGFRβ expression and showed a strong correlation between TYKRIL and PDGFRβ in PAH patient samples. In addition, TYKRIL can function as a decoy to bind strongly with p53 and regulate its activity by disrupting the formation of p53-p300 complexes. As TYKRIL does not have a mouse homologue, in order to investigate the translational potential of TYKRIL, ex vivo silencing of TYKRIL using LNA GapmeRs in viable precision-cut lung slices (PCLS) from the lung tissues of IPAH patients decreased pulmonary vascular remodeling. In addition, GapmeR inhibition of TYKRIL reduced PCNA-positive cells and increased the deoxynucleotide transferase-mediated dUTP nick end label-positive cells. Thus, this study suggests that TYKRIL may be a promising therapeutic target for PAH and hypoxia-associated PH [99].

LncRNA ANRIL is located at the CDKN2A/B locus at 9p21.3. It was first identified from familial melanoma patients with a large deletion at the CDKN2A/B locus [100]. The dysregulated expression of ANRIL is involved in cardiovascular disease, especially in atherosclerosis. For example, ANRIL promotes atherosclerosis progression through sponging miR-399-5p and further derepressing the RAS/RAF/ERK signaling pathway [101]. ANRIL acts as molecular scaffold to recruit WDR5 and HDAC3 to form a protein complex that can upregulate ROS level and promote HASMC phenotype transition, contributing to atherosclerosis [102]. However, another study found that ANRIL overexpression inhibits smooth muscle cell phenotypic switching to suppress the formation of atherosclerotic plaque through the AMP-activated protein kinase (AMPK) pathway [103]. In PH, ANRIL expression was downregulated in human PASMCs exposed to hypoxia. The downregulated ANRIL induced cells from the G0/G1 phase to the G2/M + S phase, as well as proliferation and migration [104]. This research only performed simple functional studies in PASMCs with the silencing of ANRIL. The exact role of ANRIL in PH is still largely unknown, such as expression profile in PH patients and the molecular mechanisms of ANRIL in the progression of PH.

### 2.6. LncRNAs Localized to the Nucleus

Microarray analysis was performed in pulmonary arteries in a rat MCT-PH model with and without metformin treatment. The increased expression of nuclear lncRNA NONRATT015587.2 was validated in a PH model in which upregulation was reversed by metformin treatment. Gof and Lof of NONRATT015587.2 affected proliferation, cell cycle progression and apoptosis in PASMCs through the p53 and HIF-1α signaling pathway [105], or by targeting p21 [106]. Highly conserved nuclear lncRNA lncPTSR was identified from RNA-Seq in PDGF-BB-stimulated PASMCs. LncPTSR expression was downregulated in PDGF-BB and hypoxia-induced PASMCs. Manipulation of lncPTSR regulates rat PASMCs proliferation, apoptosis, and migration. Importantly, in vivo inhibition of LncPTSR by adenovirus associated virus type 9 (AAV9)-mediated shRNA in rats showed spontaneous development of PH, with increased right ventricular systolic pressure and pulmonary vascular remodeling in the normoxic condition, which suggest that lncPTSR is the key pathological factor during the development of PH. Mechanically, lncPTSR functioned as a *cis*-acting lncRNA to repress PMCA4 (plasma membrane calcium transporting ATPase 4) expression and attenuated the intracellular Ca^2+^ efflux via the PDGF-BB mediated MEK/ERK signaling of PASMCs both in vitro and in vivo [107].

### 2.7. LncRNA in Right Ventricular in Pulmonary Hypertension

Right ventricular (RV) failure is the primary cause of death in PAH patients. RV dysfunction is a vital predictor of mortality in PAH. In the early-stage of PAH, the RV remains adapted with physiological responses to afterload with increased contractility and little or no effect on RV function. Persistent elevated RV afterload in the advanced stage leads to a maladaptive hypertrophy and other pathological features contributing to the RV failure and premature death. However, there is no effective drug treatment specifically targeting RV failure. Understanding the dysregulated signaling pathways and pathological transcriptomes associated with lncRNAs during RV remodeling in PH may provide valuable therapeutic strategies [108,109]. Analysis of long non-coding RNA and mRNA profiles in the right ventricle myocardium during acute right heart failure in PH rats revealed a set of dysregulated lncRNAs in the RV tissues [110]. LncRNA H19 was significantly upregulated in decompensated RV from PAH patients and positively correlated with RV cardiomyocyte hypertrophy and cardiac fibrosis. Consistent with the PAH patient findings, H19 expression was increased in two preclinical RV failure rat PH models (MCT and PAB (pulmonary artery banding)). In addition, the processed products of H19 (miR-675-3p and miR-675-5p) showed similar expression patterns to H19, which suggests that the H19/miR-675 axis participated in the RV remodeling/failure. GapmeR inhibition of H19 in vivo prevented cardiac remodeling by reducing hypertrophy, fibrosis and capillary rarefaction, which ameliorated RV function and increased the survival rate of two rat RV failure PH models, but without affecting the pulmonary vascular remodeling. Mechanically, H19/miR-675 promotes the maladaptive prohypertrophic growth of cardiomyocytes, partially due to downregulation of the E2F2/EZH2 axis. However, this mechanism is not applicable to H19/miR-675 promoting the proliferation and activation of cardiac fibroblasts, which suggests a cell-specific function and mechanism of lncRNA. Interestingly, H19 expression levels in plasma discriminated PAH patients from controls, correlated with RV function and long-term survival rate, in two independent IPAH patient cohorts, which suggests that plasma H19 levels are promising biomarkers in RV failure and outcomes in PAH [111].

## 3. CircRNAs

### 3.1. CircRNA Biogenesis and Function

CircRNAs are newly identified noncoding RNAs that undergoes backsplicing/head-to-tail splicing from mature RNA transcripts and form covalently closed loop rings [112]. CircRNA can be divided into exonic circRNA (EcRNA), intronic circRNA (CiRNA), and exon-intron circRNA (EIcRNA) [9]. Most circRNAs are derived from known protein-coding genes and consist of a single exon or of multiple exons [113], which occur when the 3′ end of an exon links to an exon at the same 5′ end, or an upstream exon, to form a closed RNA loop [114]. Circular intronic RNAs are intron lariats formed by excised introns, when the 5′ splice site joins with the branchpoint (BP) during splicing. CiRNAs accumulate in human cells due to escape from debranching, which regulates parent gene expression by modulating elongation Pol II activity [115]. CircRNAs are highly abundant in the human genome and generally lower expression compared with mRNAs. Similar to lncRNAs, the expression patterns of circRNAs relate to cell type, tissue-specific and developmental stages. However, the evolutionary conservation of circRNA is better than lncRNA [8,116]. CircRNAs lack of adenylation (poly(A)) and are not polyadenylated, without capping and resistance to exonucleases. Therefore, circRNA are more stable than linear RNAs [117]. Following biogenesis, most circRNAs are exported to the cytoplasm, with the exception of intron-containing circRNAs [118]. The dysregulation of circRNAs has been shown to be involved in various pathophysiological conditions and has attracted growing interest in the scientific research of cardiovascular disease [119], including PH [18].

The biological functions and mechanisms of circRNAs have only been investigated for a small fraction of the circRNAs identified to date. Approximately 10% of genes can undergo backsplicing to produce circRNAs [120]. Over 100,000 unique human circRNAs have been discovered, which are under-studied compared with other noncoding RNAs in the human transcriptome [121,122]. CircRNAs can function in *cis* by balancing the backsplicing and splicing rates of their linear mRNA counterpart [123]. In addition, circRNA can function in *trans* as ceRNAs binding to miRNAs [124], directly interacting with regulator proteins [125], regulating gene expression (e.g., epigenetic control, splicing, transcription, and translation) [126], and encoding peptides or proteins [127]. There has been an increasing focus on characterizing disease relevance and functional mechanisms of circRNA in PH due to their emerging potential as promising therapeutic targets [18]. However, our knowledge of circRNA, either in biogenesis or functional or the pathophysiological roles in the human disease, is still in its infancy. Future development of new technologies, comprehensive understanding in functional mechanisms of circRNAs, and the application of effective strategies to target circRNAs in preclinical models, will contribute to the successful development of circRNA-based therapies in PH (Figure 2).

### 3.2. CircRNA in Pulmonary Hypertension

#### 3.2.1. CircRNAs Identified from Patient Samples

Microarray was performed on whole blood samples from chronic thromboembolic pulmonary hypertension (CTEPH) patients and healthy controls, and initiated a burst in the field of circRNA research in PH. There were 351 differentially expressed circRNAs between the CTEPH and control groups. Has_circ_0002062 and has_circ_0022342 are promising circRNAs participating in the development of CTEPH [128]. Independent study further confirmed has_circ_0002062 was upregulated in PASMCs exposed to hypoxia. Silencing of has_circ_0002062 prevented cell proliferation and migration in PASMCs by acting as a ceRNA targeting miR-942-5p/CDK6 axis. In addition, CDK6 inhibition by AAV in vivo prevented pulmonary vascular remodeling in a hypoxia-induced PH model [129], which suggests that the downstream target of has_circ_0002062 shows therapeutic potential. Thus, manipulation of has_circ_0002062 may be a promising therapeutic strategy for PH. However, further studies need to address whether has_circ_0002062 has homologous circRNA in mice, and the therapeutic role of the mouse homolog, as well as the effects of other downstream targets. Another independent study performed circRNA chip-on-the-blood samples from CTEPH patients and identified upregulated has_circ_0046159 and has_circ_0026480 in the CTEPH group. Bioinformatics analysis suggested that the has_circ_0026480/miR-27a-3p/ATXN1 axis and the has_circ_0046159/miR-1226-3p/ATP2A2 axis are the molecular mechanisms involved in the pathogenesis of PH [130]. However, this study did not perform wet-lab work to further investigate the function and mechanism of the selected circRNAs.

CircRNA microarray analysis was performed on lung tissues of COPD patients with PAH. Has_circ_0016070 was the only upregulated circRNA in the COPD (+) PAH (+) group compared with the COPD (+) PAH (−) group. Inhibition of has_circ_0016070 prevented PASMC proliferation by regulation of cell cycle arrest at the G1/G0 phase by the miR-942/CCND1 axis [131]. A microarray screen in whole-blood samples of COPD-PAH patients and control groups found 158 dysregulated circRNAs in patients with COPD-PAH. Has_circNFXL1_009 was the most significantly downregulated circRNA among the validation set. Manipulation of has_circNFXL1_009 affected the expression and activity of K^+^ channels and cell proliferation, migration, and apoptosis in PASMCs through targeting the miR-29b-5p/KCNB1 axis [132]. Increased expression of circATP2B4 was found in the serum of PAH patients and hypoxia-induced PASMCs. CircATP2B4 promoted the hypoxia-induced proliferation and migration in PASMCs through the miR-223/ATR axis [133].

CircRNAs have good conservation across species, with high abundance and stability in plasma, saliva, and urine, which are ideal properties for biomarker development. Few dysregulated circRNAs in PAH have shown promising disease biomarker potential. For example, circRNA_0068481 expression was increased in right ventricular hypertrophy (RVH) of PAH patients and the expression levels of circRNA_0068481was associated with RVH severity and capable of predicting RVH in PAH patients. Mechanistically, circRNA_0068481 act as ceRNA to sponge miR-646, miR-570 and miR-885, which further release the depression of EYA3 expression [87]. In addition, the serum level of circRNA_0068481 was significantly upregulated in IPAH patients and positively correlated with the severity of PAH disease parameters, including heart function, disease risk stratification, right heart failure, and patient death [134]. These studies suggest that circRNA_0068481 is a promising noninvasive biomarker for diagnosis, stratification and clinical outcome prediction of IPAH. In addition, a lower expression level of circGSAP (circular γ-secretase activating protein) was found in PBMCs from IPAH patients and was associated with the occurrence and poor prognosis of IPAH, which is a promising biomarker for the diagnosis and prognosis of IPAH [135]. Has_circ_0003416 was identified from a previous microarray assay in plasma samples from CHD (congenital heart disease) heart and CHD-PAH patients (GSE171827). Downregulation of has_circ_0003416 was found in the plasma samples of CHD-PAH patients compared with CHD patients and healthy controls in a pediatric population, which suggest that has_circ_0003416 may serve as a diagnosing biomarker of PAH-CHD [136].

#### 3.2.2. CircRNAs Identified from PH Rodent Models

Microarray circRNA assays were performed on the lung tissues from hypoxia-induced mice in a mouse PH model, and 64 dysregulated circRNAs were identified. Bioinformatics analysis and experimental validation suggest that mmu_circRNA_004592 and mmu_circRNA_018351 are two functional important circRNAs [137]. N6-methyladenosine (m6A) modification of circRNAs participated in the pathogenesis of multiple diseases [138]. RNA-Seq of lung tissues in a hypoxia-induced rat PH model identified 166 upregulated m^6^A abundance circRNAs and 191 downregulated m6A abundance circRNAs. M^6^A levels in circRNAs were decreased in lungs of hypoxia-induced PH subjects and PASMCs/PAECs exposed to hypoxia. M^6^A circRNAs were mainly derived from single exon of protein-coding genes from rat lung tissues in control and hypoxia-induced PH subjects. In addition, circXpo6 and circTmtc3 are two novel circRNAs modified by m^6^A in hypoxia-mediated pulmonary hypertension [139]. These circRNAs are still at validation stages, and their functional and mechanistic roles in PH are still largely unknown. In contrast to lack of functional investigation, circ-calm4, alternative splicing of the calmodulin 4 was identified from the RNA-Seq of lung tissues in a hypoxia-induced PH mouse model, and expression was upregulated in the PH. Manipulation of circ-calm4 affected PASMC proliferation by regulating the cell cycle through sponging the miR-337/Myo10 axis [140]. A subsequent study found that circ-calm4 regulates hypoxia-induced pyroptosis through the circ-calm4/miR-124-3p/PDCD6 axis [141]. Importantly, both AAV5 and AAV9 mediated knockdown of circ-calm4 significantly prevented the development of PH in vivo [140,141]. Additionally, mmu_circ_0000790 expression was significantly upregulated in the lung tissues in a hypoxia-induced PH mice model and PASMCs from mice. Gof and Lof manipulation of mmu_circ_0000790 affected cell proliferation and apoptosis in PASMCs in vitro, as well as pulmonary vascular remodeling in vivo in a hypoxia-induced mouse PH model. Mechanistically, mmu_circ_0000790 participated in the pathogenesis of PH through sponging the miR-374c/FOXC1 axis, and an in vivo PH model validated the therapeutic effect of overexpression of miR-374. Thus, the mmu_circ_0000790/miR-374c/FOXC1 axis is a promising therapeutic target for PH disease [142]. CircRNA circSIRT1 expression was elevated in lung tissues in a PH rat model and PASMCs exposed to hypoxia. Silencing of circSIRT1 prevented the development of PH in vivo and suppressed PASMCs proliferation, migration and autophagy in the hypoxic condition in vitro via circSIRT1/miR-145-5p/Akt3 [143]. These above studies investigated the in vivo therapeutic effect of circRNAs and identified different molecular mechanisms and functional roles of circRNA acting as ceRNA. However, there is still the lack of investigations of human and rat homologues. PH clinical sample validation and a preclinical rat PH model investigation are key steps for translational research.

A number of circRNAs without in vivo investigation have been shown to be involved in the development of PH, and mainly function as ceRNAs to sponge miRNAs or protein. Vascular calcification has been found in a hypoxia-induced rat PH model [144]. However, the underlying mechanisms are still largely unknown. The expression of bone-related proteins during calcification and the circRNA, CDR1as, was significantly increased in PASMCs with hypoxia exposure. Functionally, CDR1as facilitated the phenotypic switch of PASMCs from a contractile to an osteogenic phenotype. The underlying mechanism of CDR1as was as a ceRNA to sponge the miR-7-5p/CNN3 and CAMK2D axis [145]. Circ_Grm1 was induced by hypoxia stimulation in PASMCs, and silencing of circ_Grm1 suppressed the proliferation and migration of PASMCs in the hypoxic condition. Mechanistically, circ_Grm1 modulated its host gene Grm1 by binding to FUS (FUS RNA binding protein) and regulation of the Rap1/ERK pathway [146]. Platelet-derived growth factor (PDGF) induced circHIPK3 upregulation and contributed to the PAEC proliferation, migration and angiogenesis through the circHIPK3/miR-328-3p/STAT3 axis [147]. Hypoxia-induced circWDR37 promoted the PASMC proliferation, migration, cell cycle progression and apoptosis resistance through sponging miR-138-5p [148]. The PH-relevant interactions of circRNA-miRNA-mRNA axis are shown in Table 1.

### 3.3. Forefront Areas of Non-Coding RNA Research

Lage amounts of dysregulated noncoding RNAs in PH have been identified. However, the functional mechanisms of the majority of noncoding RNAs are still largely unknown. Investigation of noncoding RNA (lncRNAs and circRNAs) binding partners and structures are key for understanding their biological functions and molecular mechanisms. Emerging technologies of recent years have provided powerful new approaches to address these key aspects of ncRNA studies and further uncover new potentially unexpected biological functions. PH is a highly heterogeneous disease with multifactorial and complex aetiology and with varied treatment responses [150]. In addition, lncRNAs are a heterogeneous class of molecules that have key biological functions through regulating gene expression or acting as master regulators in pathophysiological processes [151]. Thus, it would be important to investigate these cell-/tissue-/condition-specific ncRNAs in PH to develop therapeutics targeting ncRNAs, or to develop ncRNA biomarkers.

### 3.4. Single Cell and Spatial Transcriptomics

Over past decades, scientist have used bulk sequencing methods to investigate gene expression dysfunctions in different cells and tissues. Bulk RNA-Seq is widely used in identification of the average expression profiles of dysregulated lncRNAs or circRNAs from cells or tissues at the time or condition of measurement. However, there are limitations of bulk RNA-Seq application in complex tissues with heterogeneous cell types, as well as in underling dynamic processes such as development and differentiation. As pulmonary vasculature cell populations show high degrees of cellular and transcriptomic heterogeneity, bulk RNA-Seq is unable to identify the individual complexity of each cell or the heterogeneity tissues [152]. In addition, the expression profiles of numerous lncRNAs and circRNAs are thought to be specific to subpopulations of cells in the lung and right ventricle in PH. Meanwhile, the expression levels of ncRNAs are relatively low compared with mRNA in bulk transcriptomic analysis. Therefore, the true signals contributing to the pathogenesis of PH from specific cell population or cell types can be confused by the average gene expression profile from bulk RNA-Seq. These challenges for bulk RNA-seq have urged the development of single cell RNA-Seq (scRNA-Seq) techniques. Recent studies performed scRNA-Seq techniques on human lung tissues from IPAH patients [153], rat PH models and lung endothelial cells in a PH mouse model [154] and identified new dysregulated genes and pathways, and a subpopulation of cells in PH. These single-cell transcriptomic datasets generated from specifical cell types, animal PH models and human patients, have initiated the investigation of species-specific, model-specific, and subtypes of disease-specific variability in the cellular composition of lung and right ventricular in PH disease [155]. However, there are currently no published studies on ncRNA transcriptomic profiles by scRNA-Seq in PH. Thus, investigating ncRNAs expression profiles at the single cell level in the lung and right ventricle may reveal unique molecular subtypes or rare cell populations of PH, which can provide molecular insight into PH heterogeneity. However, scRNA-Seq has its limitations as well. The process of preparing single cell samples may stress the cells and alter the intracellular signals, samples preparation requires techniques expensive for most researcher, and both scRNA-Seq and bulk RNA-Seq lack critical spatial information.

Driven by the limitations of scRNA-Seq, spatial RNA sequencing (spRNASeq) is a newly developed molecular profiling technology. This technique can combine the advantages of transcriptional analysis of bulk RNA-seq and in situ hybridization, mapping where the activity is occurring and providing whole spatial transcriptomics data. The SpRNASeq technique can be used to delineate extensive spatial gene expression and single cell levels, which could compensate for the lack of spatial information in scRNA-Seq datasets. Other spatial profiling methods, such as fluorescent in situ RNA sequencing [156], barcode in situ targeted sequencing [157] and multiplex immunofluorescence staining [158], can be applied to determine the spatial distribution of cellular subpopulations, cell lineages pathways, and complex cell-cell communication pathways in the development of angioobliterative pulmonary vascular lesions and right ventricular remodeling. Together, using scRNA-Seq and spatial profiling techniques, generating unbiased spatial transcriptomic profiles of ncRNAs in PH will advance our understanding of the pathophysiological processes underlying PH. This will help the selection of promising candidates for drug development and the design of novel therapeutic strategies targeting disease-specific subpopulations of dysregulated ncRNAs in PH. Such advances in techniques will fuel future years of scientific discoveries in the PH field.

### 3.5. NcRNA-Protein Interactions

Most lncRNAs and circRNAs need to bind a variety of proteins to perform their biological functions [159].Identification of the interacting proteins of lncRNAs/circRNAs is the key step to elucidate lncRNAs/circRNAs functional and mechanistic research. Experimental methods and computational methods are the two major methods to explore ncRNA-protein interaction. There are numerous computational tools and databases to predict RNA binding proteins. Although the experimental methods are time-consuming and expensive, validation of the interacting protein is a key step to further investigate the functions and molecular mechanism of lncRNAs/circRNAs [160]. Traditional RNA pulldown methods use immobilized synthetic labeled RNA or DNA probes targeting the lncRNAs/circRNAs of interest to incubate with proteins from cell lysates to form RNA-protein complexes in vitro [161]. The in vivo approach of RNA pull-down is more complex than in vitro, which requires crosslinking RNA-protein complex to preserve RNA-protein interaction, adds modified probes to capture aptamer-tagged RNAs of interest from cells and detects the RBPs subsequently [161,162]. Although in vitro methods lack endogenous physiological and pathological conditions, they are relatively simple, highly efficient and easy to perform compare with in vivo methods. In addition, both in vivo and in vitro methods need to be coupled with mass spectrometry to identify global RNA-protein interaction profiles. For example, lncRNA Rps4l was found to bind with transcription factor ILF3 by RNA-pulldown, followed by mass spectrometry identification and Western blot validation [93]. Alternatively, computational methods can predict the interaction proteins of lncRNAs/circRNAs and then be followed by experimental validation. The major validation experimental approach for lncRNAs/circRNAs-protein interaction prediction is RIP (RNA immunoprecipitation) followed by MS (mass spectrometry), Western blot and qPCR analysis [163]. In order to comprehensively understand the protein binding partners of lncRNAs/circRNAs, as well as the specific protein binding sites in the lncRNAs/circRNAs, various in vivo and in vitro RNA pulldown methods have been developed, such as cross-linking and immunoprecipitation (CLIP) in living cells [164]. As detailed above, the majority of lncRNAs/circRNAs research has focused on sponging function, with few studies investigating their binding proteins’ function and mechanism in the pathogenesis of PH. Future studies are required to identify the binding partners of lncRNAs/circRNAs to probe the mechanisms.

### 3.6. Biomarker of LncRNAs and CircRNAs in PH

Currently, hundreds of lncRNAs/circRNAs have been found to be dysregulated in PH and some of them participate in the initiation and progression of PH. In addition, altered expression levels of several lncRNAs/circRNAs in plasma have been found to correlate with clinical features of PH. The wide application of next-generation sequencing and increased research of ncRNAs in PH will expand the numbers of PH-associated lncRNAs/circRNAs. Given the prevalence, specific expression, functional importance and therapeutic potential of lncRNAs/circRNAs, there is intense interest in evaluating lncRNAs/circRNAs as clinical biomarkers in PH. In 2012, the FDA approved the first lncRNA PCA3 serving as biomarker for prostate cancer in urine. This successful lncRNA biomarker has set the ground for future development and application of ncRNAs for the diagnostic and prognostic for PH. In addition, circRNA shows high stability and abundant in liquid biopsy, and thousands of circRNAs have been found enriched in blood samples [121]. Plasma H19 levels discriminated PAH patients from controls, correlated with RV function and predicted long-term survival in two independent IPAH cohorts, as well as in preclinical PH models, which suggests H19 could serve as a promising diagnostic and prognostic biomarker of PAH [111]. Aside from H19, there are several lncRNAs, including MALAT1 [80], CASC2 [82,83], has_circ_0003416 [136], circRNA_0068481 [149], and circATP2B4 [133], that are upregulated in the plasma of PAH patients, and circGSAP is decreased in PBMCs from IPAH patients [135]. These are in the early stage of biomarker development for the diagnosis and prognosis of PAH. However, these studies lack samples from a large cohort of patients, and samples from different stages of PAH, as well as lack the mechanisms responsible for the development of PAH. It is a long journey, and challenging work, from the experimental identification of lncRNAs/circRNAs biomarkers to translation into clinical practice. The challenges include sample collection time-point in the disease onset or progression stages, sample quality, handling and storage, RNA extraction and detection methods, lack of large cohort patient samples and ideal preclinical models, and largely unknown functions and mechanisms of lncRNAs/circRNAs in PH. Remarkably, due to the importance of lncRNAs/circRNAs in the pathogenesis of PH, prospective studies are able to develop the ncRNAs molecules as next-generation biomarkers compared with traditional protein biomarker by comprehensively investigating their expression, function, and molecular mechanisms. The development of novel biomarker will expand the understanding of PH pathologies and would enable more accurate risk stratification, diagnosis, and prognosis of PH management in clinical practice.

## 4. NcRNA Therapeutics

The revolution of next-generation deep sequencing provides an enormous database for noncoding RNAs (lncRNAs/circRNAs) associated with specific cell/tissue and individuals in various disease stages of cardiovascular disease, including PH. Many of them have been shown to involved in the regulation of homeostasis of vascular remodeling in PH and may comprise novel therapeutic targets for disease treatment. This allows the opportunity for a major breakthrough in personalized therapeutic strategies. However, the ncRNA research field in PH is still in its infancy, and the majority studies are limited to in vitro work and still in in preliminary investigations. Only a few lncRNAs/circRNAs studies so far have systematically investigated the interaction of lncRNAs/circRNAs with proteins. Thus, the diverse mechanisms of lncRNAs/circRNAs are largely unknown. LncRNAs TUG1, PAXIP1-AS, H19, and circRNAs circSIRT1, mmu_circ_0000790, circ-calm4 were found to contribute to the pathogenesis of PH in preclinical studies. Silencing of these RNA molecules prevented pulmonary vascular remodeling by inhibiting PASMCs proliferation, migration and induced apoptosis. Overexpression Rps4l by AAV9 showed a protective role in an animal PH model [94]. Given their important functional roles, manipulation of the expression levels of lncRNAs/circRNAs by loss-of-function and gain-of-function have emerged as potential novel therapeutic options. However, delivery strategies also need to be optimized.

Two major categories of oligonucleotide-based RNA therapeutic approaches are RNAi (siRNA) and antisense oligonucleotides (ASOs), which have been widely used to target both coding and noncoding RNAs. SiRNAs are RNA duplexes that are mostly chemically modified to increase the stability and limit immunogenicity. In 2018, the FDA approved the first siRNA therapeutic, Patisiran (brand name Onpattro), for familial transthyretin-mediated amyloidosis [165]. On the contrary, ASOs are single synthetic oligonucleotide strands interacting with RNA, resulting in gene silencing. In addition, antisense LNA GapmeRs are chimeric ASOs containing a central block of deoxynucleotide monomers that induce degradation by RNase H-dependent mechanisms. There are numerous antisense-based drugs under clinical development, including for cardiovascular diseases, such as PCSK9 in LDL-C-hypercholesterolemia (NCT01350960, NCT02597127) [166]. Apart from oligonucleotide-based RNA therapy, recombinant viral system can package shRNA into viral delivery tools to inhibit lncRNAs/circRNAs expression [167,168]. For example, adenovirus-associated virus type 9 (AAV9)-mediated shRNA LncPTSR in rats resulted in spontaneous development of the PH phenotype [107]. AAV5 and AAV9-mediated knockdown of circ-calm4 significantly prevented the development of PH in vivo [140,141].

The three main recombinant viral systems include adenoviruses, lentiviruses and adeno-associated virus (AAVs). Due to high transduction efficiency, no genomic integration, and robust transgene expression in infected cells, adenoviral-mediated gene delivery strategies have promising translation applications in gene therapy in human disease [169]. The AAV system, with no risk of genomic integration, is the most widely use platforms for gene delivery in scientific research and clinical trials. Multiple AAV serotypes make tissue-specific gene delivery possible both in vitro and in vivo [170]. AAV9-mediated overexpression of the Rps4l attenuated the development of mice in a PH model [94]. However, it would be difficult to package into AAV if the lncRNAs/circRNAs transcripts are over 4.7 kb, which limits the application of an AAV-based therapeutic approach. On the contrary, lentiviral vector transduction can intergrade virus DNA into the host genome, which enables long-term expression of gene fragments compared with transient transduction of adenovirus and AVVs. Lentivirus-mediated overexpression of lncRNA miR-503HG in an SU5416/hypoxia mouse PH model reduced EndMT transition in the lung [67]. Although in vitro and preclinical animal models have shown promising results by lentiviral transduction, the genomic integration issue limits the development of lentiviral-based therapeutic drugs. In addition, the function of lncRNAs/circRNAs is largely dependent on subcellular distribution. A viral vector overexpression strategy will result in the lncRNAs/circRNAs expressed both in cytoplasm and nucleus, which result in misleading functional findings.

In addition to a viral-based delivery strategy, liposomes and lipid nanoparticles (LNPs) are two main non-viral vector delivery methods by which to deliver ncRNAs. Liposomes are artificial phospholipid vesicles consisting of a bilayer structure, which can be loaded with therapeutic cargos. The size of liposomes can range from a few nanometers to several micrometers. Nanoliposomes are considered an ideal drug delivery system for the encapsulation of a wide range of therapeutics such as siRNA, mRNA, DNA, RNA and protein, having low toxicity and biological compatibility. Direct siRNA delivery may have some unfavorable properties including easy degradation, low cellular uptake, low efficiency target in nucleus, and low endosomal escapability. Thus, liposomes have been considered to be one of the most outstanding nanocarriers for drug deliver and employed as potent drug carriers in targeting ncRNAs. In addition, liposomal spherical nucleic acid (LSNA) can be used to effectively delivery ASOs to the nucleus, which can interact with the target lncRNA and cause subsequent knockdown [171]. For example, the arginine-glycine-aspartic acid peptide (RGD)-modified PEGlyated cationic liposome (RGD-Lip) is a novel gene delivery system to target the placenta. SiRNA H19x loaded into RGD-lip, was successfully transferred into the placenta of C57BL/6 mice resulting in the occurrence of preeclampsia-like symptoms by inhibiting H19x expression [172]. Intratracheal administration of liposomes selectively delivered to the lung in an MCT PH model, which could enhance vascular permeability by the inflammatory response. Thus, drug encapsulation in liposomes could be an effective therapeutics delivery strategy in PH patients [173].

LNPs are composed of multiple lipid layers, as well as microdomains of lipid and nucleic acid, and can be easily modified. Cellular uptake is mediated by endocytosis and cargo is released into the cytoplasm. LNPs are an important class of delivery system that has applications in clinical treatment. The FDA approved the siRNA therapeutic Patisirna chose LNPs delivery strategy [174], which is opening the gate of future RNAi therapeutics. mRNA vaccines for COVID-19 have been widely used in the world [175,176]. MRX34 is under a phase 1 clinical trial for advanced solid tumor treatment, and uses LNPs to deliver a double stranded miR-34a mimic [177]. SiRNA targeting lncRNA LINC01257 was loaded into LNPs, and the expression of LINC01257 was decreased with the delivery of LNP-si-LINC01257 in the cell line model of pediatric acute myeloid leukemia (AML) [178]. Oncogenic lncRNA Dancer is highly expressed in triple-negative breast cancer (TNBC). Nanoparticle mediated RNAi of DANCR prevented the progression of TNBC both in vitro and in vivo [179]. In addition, hexadecyl-treprostinil (C16TR), a prodrug of trepreostinil (TRE), formulated in LNPs for inhalation as pulmonary vasodilator, showed that C16TR-LNP provides long-acting pulmonary vasodilation [180]. In PH, LNPs delivery of an miR-145 inhibitor prevented the development of experimental PH [181]. Although LNP delivery of ncRNA therapeutics has not been widely tested in PH, the successfully delivered mRNA for COVID-19 vaccines marks an important milestone of LNP-therapeutics in human disease.

Targeting noncoding RNAs (lncRNAs/circRNAs) is regarded as a promising option to develop new-generation therapeutics for PH. Functional characterization of lncRNAs/circRNAs is mandatory for RNA-based therapeutics development. Although the number of publications on lncRNAs/circRNAs in PH field is quickly rising, most studies have not rigorously investigated the functional and mechanistic of those RNAs due to obstacles in ncRNAs/circRNAs research. First, lncRNAs/circRNAs are not well conserved across different species compared with mRNA. This makes it difficult to carry out preclinical model and translational studies. Identification of the RNA interaction partners and developing human models may be necessary to solve this problem. Second, most lncRNAs are preferentially located in the nucleus, while majority of circRNAs have a cytoplasmic localization. Research has shown that the subcellular localization of lncRNAs/circRNAs is crucial to determine whether to target them using RNAi or antisense oligonucleotides. For cytoplasmic RNA, siRNA-mediated knockdown is quite efficient, especially for in vitro work, as siRNAs function better in the cytoplasm. For nuclear-expressed and accessible RNAs, GapmeRs may be the best choice, as RNase H–dependent degradation is preferentially effective in the nucleus. Generally, naked single-stranded RNA strands need to be chemically modified to increase stability, especially for in vivo work. In addition, chemically modified antisense oligonucleotides GapmeRs show even better stability and a longer knockdown effect. GapmeRs have been widely used in targeting lncRNA in vivo [182]. For example, GapmeRs-mediated lncRNA H19 improved right ventricle function in an MCT rat model and pulmonary artery banding in a rat model in vivo [111]. GapmeRs haven’t been used so far for inhibition of circRNA in vivo [9]. However, it is conceivable that GapmeRs could inhibit circRNA as well. LncRNA has multiple splicing variants, and circRNA has its linear transcript. To avoid the off target and unspecific effects of oligonucleotides, genomic editing at the RNA locus would be the ideal choice. For example, CRISPR-Cas13 is new tool specific to RNA modulation, with Cas13a binding and degrading RNA in mammalian cells [183], which can provide validation results of oligonucleotide-based methods. Third, lncRNAs/circRNAs are generally of low expression, with tissue-specific and cell type-specific expression patterns. It is difficult to deliver either RNA inhibitors or viral vectors to some specific cell layers. For example, smooth muscles cell layers significantly contribute to the distal pulmonary vascular remodelling during PH disease progression. With current delivery technologies, we still cannot solve the problem of delivery of ASOs or siRNAs and even lentivirus vectors to the smooth muscle layer in vessels, which is a common issue for vascular disease. Therefore, the hurdles for developing RNA-based therapeutics are more about chemistry, toxicity and delivery. Recent clinical trials have shown significant advancements in the chemical modification and delivery strategies of RNA-based therapeutics, with improved target specificity, efficiency and stability, and reduced toxicity and unwanted adverse effects [184]. While most of the clinical trials of RNA-based therapeutics are focusing on protein-coding genes, the increasing focusing on functional lncRNAs/circRNAs from the in vitro and in vivo preclinical studies will provide a novel repertoire of candidates for RNA-based therapeutics in PH.

## 5. Conclusions

The field of pulmonary hypotension over the past years has resulted in no approved new drug for clinical treatment. There is no cure for PH. Current treatment only can improve the symptoms and slow the disease progress. The 5-year survival rate of PH reached more than 60% in 2015 but the mortality is still high [185]. The difficultly in developing a drug to cure PH is partially due to lack of the understanding of the underlying mechanisms of PH. The traditional therapeutic strategies are mainly targeting the coding genome. Recently, with the widely application of next-generation sequencing technology, lncRNAs/circRNAs have been found to be dysregulated in PH. The identification of functional lncRNAs/circRNAs in the pulmonary vasculature and right ventricle led to understanding the pathomechanisms of PH. Increasing evidence suggests that these RNA molecules exhibit a wide range of molecular functions and provide promising opportunities for biomarkers and therapeutic targets. However, only few lncRNAs/circRNAs have been deeply investigated in their functional and mechanistic roles in the physiopathological conditions of PH. The development of disease biomarkers and therapeutic targets of ncRNAs/circRNAs are still at the infant stage. miRNA therapeutics has been intensively developed in the past two decades but the application of miRNA therapeutics in vascular disease still facing many hurdles. The advancement of new technologies such as scRNA-seq, spatial transcriptome, modification of siRNA/ASOs, RNA-pulldown, and new in vitro/vivo delivery systems, will contribute to the understanding of the biological role of lncRNAs/circRNAs and all the development of RNA-based therapeutics for PH treatment.

## Figures and Tables

**Figure 1 biomolecules-12-00796-f001:**
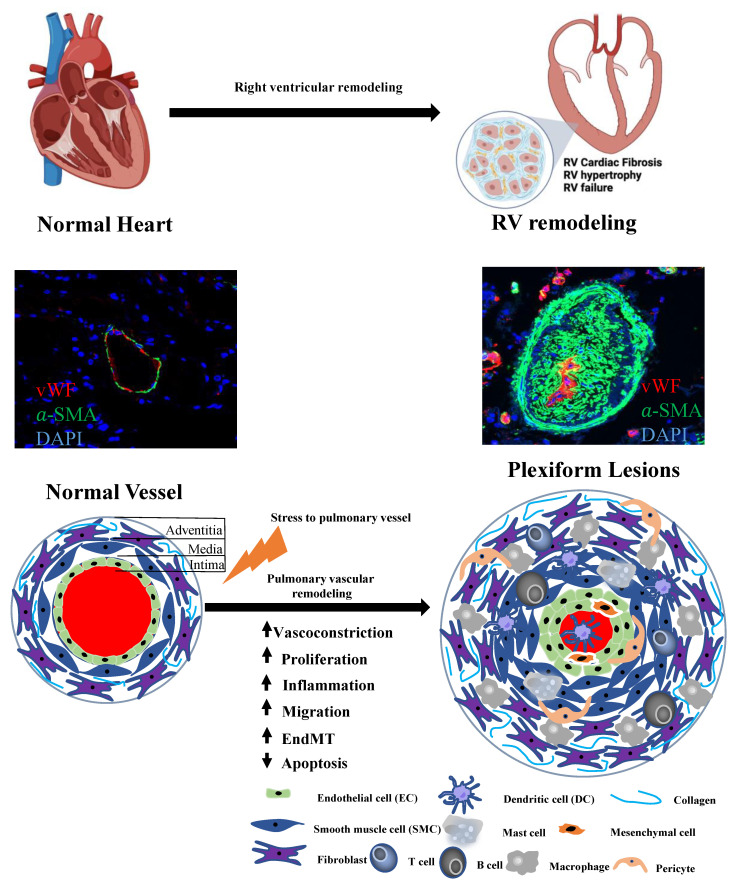
Schematic showing right ventricular and pulmonary vascular remodeling in PH. The right ventricle (RV) undergoes a remodeling process during the development of PH, which includes RV cardiac fibrosis and hypertrophy and eventually right heart failure. The pulmonary artery undergoes changes to all three layers of the vessel, including adventitial thickening with fibroblast proliferation and inflammatory and immune cell recruitment (macrophages, dendritic cells, mast cells, B cells, and T cells), proliferation, resistance to apoptosis and hypertrophy of pulmonary artery smooth muscle and endothelial cell to mesenchymal transition (EndMT), resulting in medial and intimal thickening of the pulmonary vessel wall. vWF:von Willebrand factor, α-SMA: smooth muscle alpha-actin.

**Figure 2 biomolecules-12-00796-f002:**
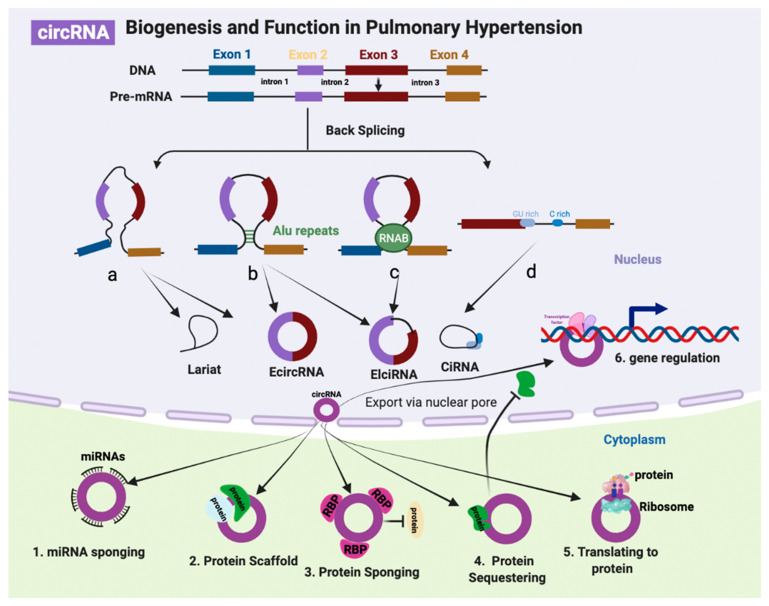
Biogenesis and functional mechanisms of circRNA. Schematic diagram showing biogenesis through canonical pre-mRNA splicing, yielding a mature mRNA molecule. (**a**) Lariat-driven circularization. (**b**,**c**) Back-splicing and circularization are medicated by RNA-binding proteins (RNAB) and Alu repeats. (**d**) ciRNA can form by some lariats removed from pre-mRNA via the canonical splicing mechanism. This process depends on a consensus motif containing a 7-nt GU-rich element near the 5′ splice site and an 11-nt-C-rich element close to the branchpoint site. Most circRNAs are exported to the cytoplasm, while few of them remain in nucleus after their biogenesis. The reported functional mechanisms include sponging miRNAs and RNA binding proteins (RBP) to decrease the binding availability of target mRNAs, assembling multiple proteins to form special protein complex, sequestering the protein in the cytosol, translating into peptide or protein, and transcriptional regulation of gene expression. The circRNAs in the diagram have been investigated in PH disease. Adapted from “circRNA in Cancer”, by BioRender.com (accessed on 25 Mar 2022) (2022). Modified from: https://app.biorender.com/biorender-templates (accessed on 25 Mar 2022).

**Table 1 biomolecules-12-00796-t001:** Selected circRNAs act as miRNA sponges and positively regulate their genes in PH.

CircRNA	Interacting miRNA	Target mRNA	Expression in PH	Functions in PH
Has-circ_0002062	miR-942-5p [129]	CDK6	Increase	Promote cell proliferation and migration
Circ_0068481	miR-646/570/885 [149]	EYA3	Increase	Biomarker of disease severity of PH
Circ-calm4	miR-337-5p [140] miR-124-3p [141]	Myo10 PDCD6	Increase	Promote cell proliferation, migration, cell cycle, and pyroptosis
Circ-CDR1	miR-7-5p [145]	CNN3-CAMK2D	Increase	Promote PASMC from contractile to osteogenic phenotype
CircSIRT1	miR-145-5p [143]	AKT3	Increase	Promote cell proliferation, migration and autophagy
Has_circ_0016070	miR-942 [131]	CCND1	Increase	Promote cell proliferation and cell cycle arrest
Has_circNFXL1_009	miR-29b-5p [132]	KCNB1	Decrease	Inhibition cell proliferation and migration; induce apoptosis
CircATP2B4	miR-223 [133]	ATR	Increase	Promote cell proliferation and migration
CircWDR37	miR-138-5p [148]		Increase	Promote cell proliferation, migration, cell cycle; apoptosis resistance
Mmu_circ_0000790	miR-373c [142]	FOXC1	Increase	Promote cell proliferation, apoptosis resistance
CircHIPK3	miR-328-3p [147]	STAT3	Increase	Inhibit proliferation, migration, angiogenesis
Has_circ_0026480	miR-27a-3p [130]	ATXN1	Decrease	
Has_circ_0046159	miR-1226-3p [130]	ATA2A2	Increase	

## Data Availability

Not applicable.

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
