# Peer review of "The Landscape of Noncoding RNA in Pulmonary Hypertension"

_biomolecules, 2022, doi:10.3390/biom12060796_

Round 1
Reviewer 1 Report
This is an interesting review on an important topic, the transcriptomics of pulmonary hypertension, devoting special interest to lncRNAs. The review is timely, complete and addresses the discussion of most of the RNAs of interest. Nevertheless, it has plenty of room for improvement since the writing is sometimes sloppy and the general layout is substandard for a scientific manuscript.
-The most important criticism is that the present manuscript is too general and poor attention is given to pathophysiological mechanisms mediated by specific lncRNAs, e.g. there’s no mention of Anril and its involvement in atherosclerosis and CVDs.
-Authors should include some information on “Pol-III-dependent Alu transcripts” , a class of ncRNAs by themselves
-For the sake of clarity, the discussion on “CircRNA in Pulmonary Hypertension” should benefit of adding a table with the details of the circRNAs better tan discussing these in the text.
- I would suggest authors to improve the section dedicated to RNA.seq since this is a quickly evolving topic and it is difficult to be knowledgeable on the latest technical developments
- In the section about ncRNA therapeutics, the authors should include a discussion on the non-viral vehicles used to deliver RNAs to the organism
-In Figure 2 there’s no correlation among the exon colors in the “gene” and the circRNAs
- There’s a double referencing system, by number and by author/year in the text (see lines 39 and 40). This should be corrected.
-Abbreviations should be standardized. Pulmonary hypertension is referred either as PH or as PAH. Furthermore, the manuscript would benefit from an abbreviations section.
-Are the figures original work? Were these got from internet? If this last is the case, are they copyright free?
Lastly, this should be carefully revised by an English native person.
Author Response
We thanks the reviewers' kindly comments and suggestions. We have addressed all the comments point by point. We hope the reviewer agree with this revision manuscript. Thank very much.

Reviewer 2 Report
In this review Deng et al try to describe the role of non-coding RNAs in pulmonary hypertension.
Overall the review is very poorly written, to the extent that sometimes it is impossible to understand the message that the authors were trying to convey. I would suggest to the authors to hire a scientific editor that can support the full re-writing of the manuscript. There are so many misspellings that I wonder if these represent lack of English knowledge or just lack of attention from the authors.
In addition, several passages in the manuscript are very descriptive and they really represent only a list of examples without any attempt from the authors to include their input/conclusion or to form any sort of narrative (for example lines 152-161, lines 210-251 etc). I believe it will be very difficult for any reader to follow the logical flow of these paragraphs.
The second part of the review is slightly better and I found particularly informative the sections regarding:
Single cell and spatial transcriptomics
ncRNA and protein interactions
Biomarker of lncRNAs and circRNAs in PH
Below is a very partial list of examples (only up to line 250 of the manuscript) in which the sentences need to be restructured and the English corrected:
Line 29 “The majority of the human genome contains is transcribed, however, only ~2% protein coding exons”. Maybe the authors meant: The majority of the human genome is transcribed, however it contains only 2% of protein-coding exons
Line 44 “ventricular hypertrophy and often died with right ventricular failure”. What does it mean “often died”?
Line 50 Please define PAH
Line 68 “Before the genomic era without the development of high-throughput sequence technologies, majority of the non-coding part of the human genome has been considered as to be transcriptional noise”. The grammar needs to be corrected.
Line 76 “lncRNAs still largely unknown”. Where is the verb?
Line 77 “Investigations the functions of lncRNA will provide”. Maybe the authors meant “investigating the function” or “investigations of the function”
Line 86 “which under splicing”. Do the authors mean undergo splicing?
Line 92 “The function roles” . Is this supposed to be “functional roles”
Line 118 “dyregulated”. Correct to dysregulated
Line 130 “Another example of LncRNA condition- and cell-specific expression of MANTIS is highly expression in endothelial cells”. This sentence makes no sense.
163 “and further lncRNA microarray”. Remove further?
180 “most of which are lack of in vivo validation”. Did the authors mean “most of which lack in vivo validation”?
204 “scanfolds”. Please correct the misspelling
In this form the paper is not suitable for publication in biomolecules.
Author Response

(The authors gave the same response as above.)

Reviewer 3 Report
Reviewer Comments:
Lin Deng and coworkers present the manuscript entitled “The Landscape of noncoding RNA in pulmonary hypertension”.
Major Points:
My main concern is because there are several recent published reviews that discuss lncRNAs and circRNAs in pulmonary hypertension. Therefore, it is not clear to me what the main contribution of this manuscript is.
Wang, Q., et al. (2022). Circular RNAs in pulmonary hypertension: Emerging biological concepts and potential mechanism. Animal models and experimental medicine, 5(1), 38–47. https://doi.org/10.1002/ame2.12208
Zang, H., et al. (2021). Non-Coding RNA Networks in Pulmonary Hypertension. Frontiers in genetics, 12, 703860. https://doi.org/10.3389/fgene.2021.703860
Jin, Q., et al. (2020). Long noncoding RNAs: emerging roles in pulmonary hypertension. Heart failure reviews, 25(5), 795–815. https://doi.org/10.1007/s10741-019-09866-2
Zhang, Y.,et al. (2019). Linking lncRNAs to regulation, pathogenesis, and diagnosis of pulmonary hypertension. Critical reviews in clinical laboratory sciences, 1–15. Advance online publication. https://doi.org/10.1080/10408363.2019.1688760
In addition, the manuscript must be proofread thoroughly to correct several syntax and grammar errors found throughout the text.
Why did the authors decide to include only the figure of circRNA biogenesis and functional mechanisms? It would be convenient for the authors to include in the figure the biogenesis of lncRNAs and their functional mechanisms, because in this review, they focus on these two ncRNAs.
Authors should include the additional sections: i) Coregulation networks of lncRNAs/miRNAs in pulmonary hypertension and ii) Coregulation networks of circRNAs/miRNAs and circRNAs/protein in pulmonary hypertension.
The authors should explain in the manuscript what is the main function of lncRNAs located in the nucleus, cytoplasm or both compartments.
The authors should include the biogenesis of intronic circRNAs formed by intron lariat.
Authors should reorganize the manuscript, there is no continuity in the text.
Authors should include a figure summarizing studies of lncRNAs and circRNAs and their functional mechanisms in pulmonary hypertension.
Minor comments:
line 12 "heterogenous" should be "heterogeneous".
Line 31 “(sncRNAs and lncRNAs)” should be “(sncRNAs and lncRNAs, respectively)”
Author Response

(The authors gave the same response as above.)

Round 2
Reviewer 1 Report
The authors have addressed my previous concerns and the manuscript now deserves its publication in Biomolecules
Author Response
We thank the reviewers' agreement for publication
Reviewer 2 Report
The authors have substantially improved the writing of the manuscript, which is now much easier to read and interpret. The content is clear and of interest to the filed.
There are however still some minor english mistakes that need to be taken care of before the manuscript can be accepted for publication.
I report below a few examples, although there are many more points throughout the text that need to be revised:
Line 169 Generally, nucleus lncRNA function as regulator=> nuclear lncRNAs
Line 172 to achieve regulation of mRNA[29].Which can block => regulation of mRNA, which can block
Line 176 processes. The gain-of-176 function and loss-of-function of lncRNAs ha been implicated in human diseases such=> have been
Line 326 The vascular endothelial cells (ECs)are located at the innermost layer of the blood vessel and directly exposure to the blood flow 326 and various stimulus=> and are directly exposed to
Line 336 molecular mechanisms and biological funcitons of lncRNA=> functions
Line 426 34a-5p[60]. Althourgh there are many 426 lncRNAs act as ceRNA participated in the pathogenesis of PH,=> although …the rest of the sentence is not clear
Line 430 There are additional lncRNAs have been implicated in the regulation=> that have been implicated
line 685 es. As 685 TYKRIL do not have mouse homologue. In order to investigate the translational potential of TYKRIL, ex vivo silen=> TYKRIL does not have a mouse homologue. Therefore, in order to investigate
Author Response
The authors have substantially improved the writing of the manuscript, which is now much easier to read and interpret. The content is clear and of interest to the filed.
There are however still some minor english mistakes that need to be taken care of before the manuscript can be accepted for publication.
Authors responses: we thank reviewers’ comments, we have carefully re-checked the English and grammar again.
I report below a few examples, although there are many more points throughout the text that need to be revised:
Line 169 Generally, nucleus lncRNA function as regulator=> nuclear lncRNAs
Authors responses: we thank reviewers’ pickup this error, we have changed correctly.
Line 172 to achieve regulation of mRNA[29].Which can block => regulation of mRNA, which can block
Authors responses: we thank reviewers’ pickup this error, we have changed correctly.
Line 176 processes. The gain-of-176 function and loss-of-function of lncRNAs ha been implicated in human diseases such=> have been
Authors responses: we thank reviewers’ pickup this error, we have changed correctly.
Line 326 The vascular endothelial cells (ECs)are located at the innermost layer of the blood vessel and directly exposure to the blood flow 326 and various stimulus=> and are directly exposed to
Authors responses: we thank reviewers’ pickup this error, we have changed correctly.
Line 336 molecular mechanisms and biological funcitons of lncRNA=> functions
Authors responses: we thank reviewers’ pickup this error, we have changed correctly.
Line 426 34a-5p[60]. Althourgh there are many 426 lncRNAs act as ceRNA participated in the pathogenesis of PH,=> although …the rest of the sentence is not clear
Authors responses: we thank reviewers’ pickup this error, we have changed correctly.
Line 430 There are additional lncRNAs have been implicated in the regulation=> that have been implicated
Authors responses: we thank reviewers’ pickup this error, we have changed correctly.
line 685 es. As 685 TYKRIL do not have mouse homologue. In order to investigate the translational potential of TYKRIL, ex vivo silen=> TYKRIL does not have a mouse homologue. Therefore, in order to investigate
Authors responses: we thank reviewers’ pickup this error, we have changed correctly.
Reviewer 3 Report
Authors have replied all my concerns, thus I suggest to accept the manuscript for publication in Biomolecules in its actual form.
Author Response

(The authors gave the same response as above.)
